# Pre-Germinative Treatments and Morphophysiological Traits in *Enterolobium cyclocarpum* and *Piscidia piscipula* (Fabaceae) from the Yucatan Peninsula, Mexico

**DOI:** 10.3390/plants11212844

**Published:** 2022-10-26

**Authors:** Thomas Martín Arceo-Gómez, Erika Robles-Díaz, Mayra D. Manrique-Ortega, Ángel Roberto Martínez-Campos, José Luis Aragón-Gastélum, Francisco Javier Aguirre-Crespo, Jorge E. Ramírez-Albores, Marlín Pérez-Suárez, Rafael Robles, Javier Reyes-Trujeque, Aarón A. Can-Estrada, Eduardo J. Gutiérrez-Alcántara, Bardo H. Sánchez-Soto, Pedro Zamora-Crescencio

**Affiliations:** 1Facultad de Ciencias Químico-Biológicas, Universidad Autónoma de Campeche, Predio s/n Avenida Ing. Humberto Lanz Cárdenas y Fraccionamiento, Ecológico Ambiental Siglo XXIII, Colonia Ex Hacienda Kalá, Campeche C.P. 24085, Mexico; 2Centro de Investigación en Corrosión, Universidad Autónoma de Campeche, Av. Héroes de Nacozari No. 480, Campus 6 de Investigaciones, Campeche C.P. 24070, Mexico; 3Instituto de Ciencias Agropecuarias y Rurales, Universidad Autónoma del Estado de México, Carretera Toluca-Ixtlahuaca km 15.5, El Cerrillo-Piedras Blancas, Toluca de Lerdo C.P. 50295, Mexico; 4Departamento de Botánica, Universidad Autónoma Agraria Antonio Narro, Calzada Antonio Narro No. 1923, Colonia Buenavista, Saltillo C.P. 25315, Mexico; 5Departamento de Ciencias Naturales y Exactas, Universidad Autónoma de Occidente, Unidad Regional los Mochis, Boulevard Macario Gaxiola y Carretera Internacional México 15, Los Mochis C.P. 81223, Mexico; 6Centro de Investigaciones Históricas y Sociales, Universidad Autónoma de Campeche, Avenida Agustín Melgar s/n entre Calle 20 y Juan de la Barrera, Colonia Buenavista, Campeche C.P. 24039, Mexico

**Keywords:** Fabaceae, germinative pre-treatments, physical dormancy, seed germination

## Abstract

*Enterolobium cyclocarpum* and *Piscidia piscipula* are two important tree Fabaceae species distributed from the Yucatan Peninsula, Mexico. Our aims were focused on the *E. cyclocarpum* and *P. piscipula* seeds for: (1) to examine the seed permeability and imbibition rate, (2) to evaluate the effect of seed pre-germinative treatments, and (3) to characterize the structures involved on the presence of physical dormancy (PY). We used fresh seeds to determine seed permeability and imbibition rate, seed viability by means of tetrazolium test, furthermore, we applied mechanical scarification and boiler shocks for 5 s, 10 s and 15 s treatments. Morphological characterization of the seed coat was by Scanning Electron Microscope (SEM). Seed viability in *E. cyclocarpum* and *P. piscipula* were 100% and 96%, respectively. Seed permeability and imbibition rate in *E. cyclocarpum* were low. The highest germination in *E. cyclocarpum* was in the mechanical scarification (92%), while in *P. piscipula*, this parameter was in the 10 s boiling water treatment (76.0%). The presence of PY was confirmed in both species because they showed low seed permeability, and imbibition rate; furthermore, exhibited macrosclereids cells. The present research seeks to promote the sustainable use of *E. cyclocarpum* and *P. piscipula*.

## 1. Introduction

Dormancy is the inability of the seeds to germinate, which can be a problem in the seedling production systems [1]. Seed dormancy is the state or condition where seeds cannot germinate even when they are under favorable environmental conditions for it to occur., including temperature, water, light, gas interchange, seed coats, and other mechanical restrictions [1]. One of the most common types of dormancies in plant species is physical dormancy (PY), in which seeds or fruit coats are water-impermeable and unable to imbibe water, limiting the germination process to be carried out [1]. To break PY, the application of germinative pre-treatments to seeds are necessary [2], e.g., heat or chemical treatments and other kinds of scarifications [2], which contribute to eliminating of the limitations that seeds have so that the germination process to be triggered.

The seeds or fruit coat are water-impermeable due to show one or more layers of palisade cells called macrosclereids [1,2,3,4]. In addition, the seed/fruit acquires water-permeability by opening of a small specialized anatomical structure within of the seed coat described as the water gap [1]. The water gap closes during maturation drying and it will open in response to environmental signals, such as humidity, fire, or high temperature, being the initial via of water entry into the seed [4]. When the water gap opens, then, the PY breaks. In the case of some Fabaceae species, the cushion-like structure is also involved in the process of seed imbibition [5,6]. Germination process is a crucial phase in plant development influenced in part by water [7]. In seeds, generally, water enters through the micropyle, and when root growth, it is the first visual sign of germination [8]. This process comprises three phases: (1) Imbibition: characterized by an intense absorption of water by the seed tissues and an increase in respiratory activity and is crucial for germination to occur [9,10]; (2) Germination: which represents the metabolic transformations necessary to the development of the seedling. In this phase, water absorption is reduced in seed and can sometimes even stop [9]; and (3) Growth: is associated with the emergence of the radicle; in this last phase, water absorption and respiratory activity increase again [8]. Particularly, the imbibition rate can be measured as the proportion of water uptake by the seeds when they are introduced into water and is considered as an indicator of the presence of PY in plant species [11]. Low imbibition rate in seeds is caused by the impermeable seed coats preventing germination. Therefore, knowledge of germination mechanisms plays a determining role in understanding the processes of propagation and regeneration of plant communities; however, very little has been studied on the seed biology in tropical plant species [12].

The Fabaceae family, commonly known as legumes, is one of the most important plant families worldwide [13]. Fabaceae is the third-largest group of flowering plants, with 19,400 species included in 730 genera, accounting for 9.4% of all flowering plant species of planet [14]. Fabaceae is the second most important food group for humans and animals, after the Poaceae [15]. In addition, the uses of these tree species can be diverse, such as food, medicinal, living fences, shade, timber, fuel, and reforestation. In several Fabaceae species have been documented the existence of PY, see [1].

In Mexico, the Fabaceae family has about 155 genera and 1903 species [16]. In the Mesoamerican region that encompasses the Yucatan Peninsula, the Fabaceae predominate in the number of species, surpassing the Poaceae and Asteraceae; this family has about 78 genera and 228 species, 21 of them endemic [17]. Fabaceae species are a representative component of the region’s landscape and are abundant in all vegetation types in the Yucatan Peninsula [18]. *Enterolobium cyclocarpum* and *Piscidia piscipula* are two representative Fabaceae species naturally distributed in the Yucatan Peninsula, which play a significant ecological and socio-cultural role.

Studies focused on evaluating the germination capacity of both species are scarce. However, the presence of PY and water-impermeability of the seeds in both *E. cyclocarpum* [19], and *P. piscipula* [12] have been suggested; nevertheless, morphological studies that support this assumption in both species are scarce. For *E. cyclocarpum*, seed pre-germinative treatments with concentrated sulfuric acid (H_2_SO_4_) obtained the highest seed germination (≥90%) [19]. However, to apply this methodology on a large scale, for example, in reforestation programs or sustainable use of this species, could have drawbacks due to the high costs and complexities to manipulate H_2_SO_4_. Thus, it is necessary to evaluate alternatives to seed germination pretreatments in *E. cyclocarpum*. In the case of *P. piscipula*, there have been very few studies on seed germination pretreatments, see González-Valdivia [20]. Therefore, in both species, the existence of PY is not clear, there are no evaluations of the seed imbibition rate, as well as the structures involved in the presence of PY (e.g., cells of macrosclereids, water gap and/or cushion-like structure).

Therefore, following hypotheses were tested in this study: (1) the seeds of *E. cyclocarpum* and *P. piscipula* will display PY, and they will increase the seed germination by applying pre-germinative treatments, and (2) the presence of PY in both species is because they show water-impermeability and low imbibition rate due to specialized cells such as macrosclereids, and structures such as water gap or cushion-like structure are involved in the process of seed imbibition in these species. Thus, we evaluated the efficiency of some seed pre-germinative treatments in *E. cyclocarpum* and *P. piscipula*; in addition, we examined the seed permeability and imbibition rate in both species, and we characterized the external and internal morphology of the testa in the *E. cyclocarpum* and *P. piscipula* seeds to describe the structures involved on the presence of PY in these species.

## 2. Results

### 2.1. Seed Viability

There were no significant differences (F (1, 98) = 2.04, *p* = 0.16) in the seed viability of the species studied. This variable showed high values and was similar for both species, 100% for *E. cyclocarpum* and 96% ± 0.24% for *P. piscipula*.

### 2.2. Imbibed Seeds and Imbibition Rate

We found no significant differences in the fresh weight for either target species over time. Mean values in *E. cyclocarpum* were 0.88 g ± 0.015 g (F (3, 196) = 0.63, *p* = 0.6), while *P. piscipula* showed mean values of 0.019 g ± 0.0003 g (F (3, 196) = 0.63, *p* = 0.6). *P. piscipula* and *E. cyclocarpum* showed relatively constant values in fresh weight during the experimentation time (72 h). Both species had values of imbibed seeds ≤4%. In the case of imbibition rate, there were also no significant differences in this variable between species (F (2, 196) = 0.99, *p* = 0.32) and species-time interaction (F (2, 196) = 1.10, *p* = 0.33). The imbibition rate in *E. cyclocarpum* was 1.49% ± 1.15%, while *P. piscipula* was 3.36% ± 1.55%.

### 2.3. Seed Germination

Regardless of the pre-germinative treatments, *E. cyclocarpum* and *P. piscipula* had final germination of 56.4% ± 4.2% and 43.2% ± 4.64%, respectively. This variable was statistically significant between species (F (1, 98) = 4.45, *p* = 0.37). Considering the pre-germinative treatments, we also found significant differences between the species (F (4, 90) = 13.79, *p* = 0.0003), treatment (F (4, 90) = 35.14, *p* < 0.0001) and the interaction between both factors (F (4, 90) = 18.24, *p* < 0.0001). The highest germination in *E. cyclocarpum* was 92% ± 4.16% in the mechanical scarification treatment, while the lowest germination was 9.0% ± 2.77% in the control. In *P. piscipula*, the highest germination was 76.0% ± 7.92% in the 10 s boiling water treatment, while the lowest was 11.0% ± 7.95% in the 5 s boiling water treatment (Figure 1).

In terms of daily germination, there were no significant differences between species (F (112, 2520) = 0.7, *p* = 0.40); however, we found significant effects in treatment (F (112, 2520) = 29.3, *p* < 0.0001), species-treatment interaction (F (112, 2520) = 13.97, *p* < 0.0001), time (F (112, 2520) = 252.41, *p* < 0.0001), the interaction between species and time (F (112, 2520) = 17.01, *p* < 0.0001), the interaction between treatment and time (F (112, 2520) = 12.82, *p* < 0.0001) and the interaction between species, time and treatment (F (112, 2520) = 9.17, *p* < 0.0001). For *E. cyclocarpum*, the highest and faster germination was documented in the treatments of boiling water for 5 s, 10 s, and 15 s, as well as mechanical scarification (35% ± 3%, 35% ± 3%, 36% ± 3%, and 37% ± 4% respectively) from the first eight days until the end of the experiment (Figure 2a).

The highest germination (92% ± 4%) was achieved in the mechanical scarification treatment from the 27th day of the experiment. In *P. piscipula*, the highest and faster germination percentages was documented in the boiling water treatments for 10 s and 15 s (41% ± 5%, 42% ± 6%, respectively) from the first four days until the end of the experiment (Figure 2b). The highest germination percentage (76.0% ± 7.92%) was reached in the boiling water treatment for 10 s from the 17th day of the experiment.

### 2.4. Morphological Characterization of the Seed Coat

#### 2.4.1. *Piscidia piscipula*

Figure 3A shows the external section of the *P. piscipula* seed where the hilar region and raphe are observed. We found the following structures within the hilar region (internal section): (1) the micropyle, which was observed as a depression located above the hilum, and (2) the hilum, which was found in the apical portion showing a circular shape and covered by remnants of funicular parenchyma with a closed central canal (Figure 3B). The seed coat of *P. piscipula* is composed of the epidermis, hypodermis, and inner parenchyma. The epidermis is composed of a layer of palisade macrosclereids or Malpighian cells. The hypodermis is composed of osteosclereids, or clock cells separated by intercellular spaces (we did not observe these cells in the hilar region). The Inner parenchyma is composed of several layers of collapsed parenchyma, possibly remnants of the endosperm (Figure 3C). In the *P. piscipula* hilar region, we also found the presence of a cushion-like structure, a tracheid bar composed of tracheids, as well as a counter-palisade layer (Figure 3D–F).

#### 2.4.2. *Enterolobium cyclocarpum*

In the external section of the *E. cyclocarpum* seed, the hilar region and lens are observed (Figure 4A). We found the following structures within the hilar region: (1) the micropyle: which was observed as a depression located to one side of the hilum showing parenchyma around, and (2) the hilum: found in the apical portion with asymmetrical shape covered with remnants of funicular parenchyma (Figure 4B). Inside the hilar region, it was observed to be composed mainly of spongy parenchyma under the hilum followed by palisade parenchyma (Figure 4C), The seed coat of *E. cyclocarpum* is composed of cuticles followed by macrosclereids, spongy parenchyma and palisade parenchyma (Figure 4D).

## 3. Discussion

We hypothesize that (1) the seeds of *E. cyclocarpum* and *P. piscipula* will display PY and they will increase the seed germination by applying pre-germinative treatments, and (2) the presence of PY in both species is because they show water-impermeability and low imbibition rate due to the presence of specialized cells known as macrosclereids, and structures such as water gap or cushion-like structure are involved in the process of seed imbibition. in this study, *E. cyclocarpum* and *P. piscipula* seeds showed high water impermeability and low seed inhibition and they showed almost all the morphological structures described above; however, the presence of PY was corroborated in both target species. In addition, we found high seed viability values in these species and consequently, the *E. cyclocarpum* and *P. piscipula* seeds became permeable and showed a high seed germination after the application of pre-germination treatments.

Water-impermeability in *E. cyclocarpum* and *P. piscipula*, showed low values (1.49% ± 1.15% and 3.36% ± 1.55%, respectively). and the percentage of imbibed seeds was also low (≤4%). In species of the genus *Vachellia* belonging to Fabaceae, Burrows [21] reported differential responses in the percentage of imbibed seeds for 48 Australian species. In terms of imbibition rate (fresh weight gain through time), Galíndez [22] documented similar low imbibition in two populations of *Amburana cearensis* (4.6% ± 3.13% and 6.6% ± 2.81% respectively), but high imbibition values in *Myroxylon peruiferum* (81.4% ± 4.01% and 95.2% ± 3% respectively). It has been suggested that the impermeability and low imbibition of seeds are indicators of the presence of PY in plant species such as Fabaceae [4,11]. Our findings about of the impermeability and low imbibition of seeds in study species support this premise.

In several species of Fabaceae have been documented the existence of PY, see [3]. Thus, the germinative responses to the application of several kinds of pre-germinative treatments e.g., [23,24,25,26,27,28,29,30] have been widely evaluated in these species. In this context, Robles-Díaz [2] documented a high germination percentage in *Lupinus rotundiflorus* (69.8% ± 2.7%) using thermal shocks with boiling water for 10 s. We found similar results because *P.*
*piscipula* had the highest germination (76.0% ± 7.95%) in thermal shocks with boiling water for 10 s. In addition, Galíndez [22] reported that mechanical scarification in *A. cearensis* is an efficient method to increase germination percentage and germination time. These findings also agree with our results because *E. cyclocarpum* showed the highest germination percentage (92% ± 4%) in the mechanical scarification treatment.

In both *E. cyclocarpum* and *P. piscipula*, research in terms of seed germination is still scarce. However, Viveros-Viveros [19] indicated that *E. cyclocarpum* showed a germination of 83% by prior immersion the seeds in 98% sulfuric acid (H_2_SO_4_) for 30 min. Similarly, Ezenwa [31] and Hernández [32] documented a high germination (93% and 92%, respectively) in H_2_SO_4_-treated (30 min and 35 min of immersion) seeds. As described above, the highest germination of the *E. cyclocarpum* seeds documented here was in the mechanical scarification treatment (92% ± 4%), which was similar to these studies.

*Enterolobium cyclocarpum* is a species widely used as shade and feed for cattle [33], and reforestation programs [34]. Furthermore, chemical entities such as terpene, waxes, resinic acids, stilbenoids, fatty acids, pectins, phenolic complexes, proteins, lignan saponins, essential oils, flavonoids, glycosides, gums (fatty acids, alkaloids, phenylpropanoids, terpenes) obtained from leaves, fruits, and logs of this species are used in the textile, pharmacological, cosmetic, and culinary industries [35,36,37]. Therefore, practical, and effective methods are needed to increase germination percentage and seedling production in *E. cyclocarpum*. In this sense, the application of H_2_SO_4_ promotes high germination percentage [19,31,32]; however, the large-scale application of this methodology may be limited, especially in the rural zones, because substantial amounts of acid are required, it could contaminate the environment, and it is expensive. Our work proposes the use of mechanical scarification as an effective method that meets the requirements described above for reaching a adequate germination.

Similarly, *P. piscipula* is a species widely used and it has diverse biotechnological approaches. Ethnomedical data showed that leaves of *P. piscipula* are used for cough, gastrointestinal, respiratory disorders [38], and aquaculture [39]. Pharmacological studies showed that *P. piscipula* induces antimycotic effects [40]. Non-polar and polar extracts from leaves exert antimicrobial activities against *Giardia duodenalis* and *Helicobacter pylori* [38]. In germinability of *P. piscipula*, González-Valdivia [20] obtained a germination of 55% through an immersion treatment in water at 100 °C for 3 min. Our results were similar because *P. piscipula* seeds had higher germination (76% ± 5% and 68% ± 5%) in thermal shocks with boiling water for 10 s and 15 s, respectively. These findings confirmed that heat shock treatments promote high germination percentage in *P. piscipula*.

It has been suggested that the *P. piscipula* [12] and *E. cyclocarpum* [19] seeds show PY; nevertheless, morphological studies that support this assumption in both species are limited. In *E. cyclocarpum*, Hernández-Epigmenio [41] corroborated the presence of PY in this species because it showed inside seeds macrosclereids and osteosclereids cells, as well as spongy parenchyma and palisade parenchyma. Robles-Díaz [2] also found these structures in Mexican *Lupinus* species. In our study, both *P. piscipula* and *E. cyclocarpum* showed similar morphological structures; however, the presence of a cushion-like structure and tracheid bar were only corroborated in *P. piscipula*. Cushion-like structure has been suggested to be found at Faboideae, a subfamily within Fabaceae, which groups *P. piscipula* [42]. *E. cyclocarpum* is found into Mimosoideae subfamily. It is possible that this is the explanation why cushion-like structure was not found in this species.

In this context, Robles-Díaz [2,6] and Perissé and Planchuelo [5] documented the presence of cushion-like structure and tracheid bars as direct evidence of the existence of PY in Mexican and Argentine *Lupinus* species respectively. Although we found almost all these morphological structures described above, our results about morphological characterization also confirm the assumption of the existence of PY in *E. cyclocarpum* and *P. piscipula* [12,19]. In Fabaceae species, differential water gap regions can be distinguished [3]. In this study, an external and internal characterization was carried out; nevertheless, the water gap in both target species is not clear.

It has been documented that the seed coat functions as regulator in imbibition phase [43] and a high and fast germination are inversely correlated with high seed coat hardness [44]. Therefore, when the seed coat is scarified, it can potentially decrease mechanical resistance to germination [45]. Thus, the thermal shocks with boiling water for 10 s (*P. piscipula*) and mechanic scarification (*E. cyclocarpum*) treatments influenced the breaking of PY, increasing the water uptake, and consequently high germination in both species.

The effect of temperature on the natural decomposition of impermeable seed coats may be through the heating effect of solar radiation on the surface layers of the dry and/or moist soils, together with night cooling, resulting in a combination of exposure to temperature fluctuations, thus favoring germination to take place [2]. In the case of *P. piscipula*, the higher germinative efficiency promoted for heat shocks by 10 s possibly favored the formation of cracks on the seeds allowing a more effective separation of the macrosclereids cells as well as of parenchymal tissue. It is possible that mechanical scarification had high efficiency in terms of germination in *E. cyclocarpum* because seeds of this species displayed a high proportion of spongy parenchyma and palisade parenchyma than *P. piscipula*, and this treatment facilitated their removal to overcome this barrier.

Our research provides new evidence about morphophysiological features of the *P. piscipula* and *E. cyclocarpum* seeds and suggests the use of effective and practical methods to potentiate the propagation of these species. In future works, it is important to characterize the initial site of water entry in the *P. piscipula* and *E. cyclocarpum* seeds; in addition, the possible external changes of seed coat after application of pre-germinative treatments and the effect of some microorganisms such as fungi species, which also could help to break the PY in these species. An aspect that has been almost neglected is the dynamics of the soil seed bank in both species, see [46,47]. Thus, studies of soil seed bank dynamics in *E. cyclocarpum* and *P. piscipula* are also crucial in future research.

## 4. Materials and Methods

### 4.1. Study Species

*Enterolobium cyclocarpum* (Jacq.) Griseb (1860), in Yucatan Peninsula is commonly known as Pich, is a tree that reaches 20 to 30 m tall, with a straight trunk and sometimes with small aerial roots at its base. Hemispherical cup, sometimes wider than tall; smooth to grainy bark. Leaves are composed of small leaflets, green flowers. Its fruits are flattened and coiled, woody pods with a shiny dark brown color, sweet smell and taste, and numerous seeds [48]. Its seeds are oval, flattened, large (1.5 to 2 cm long and 1 cm wide), brown, and with a very hard testa [19]. *E. cyclocarpum* is found in deciduous forests from southwestern Mexico to northern South America (Venezuela and Brazil) [19]. On the Gulf of Mexico, *E. cyclocarpum* is found from southern Tamaulipas to the Yucatan Peninsula and in the Pacific Ocean from Sinaloa to Chiapas. This species is generally found in disturbed areas in high-evergreen and medium sub-evergreen forests [48]. It is a representative floristic component of the semi-deciduous forest [49]. *E. cyclocarpum* is used as food or as shade for cattle [34], and it is used in reforestation programs in the Yucatan Peninsula [34]. Its flowering season goes from February to June, followed by a fruiting season between April and July [50].

*Piscidia piscipula* (L.) Sarg. (1891), in Mexico is commonly known as Jabín [33], is a tree tall (≥20 m), has a dense crown, fissured bark, ovate compound imparipinnate leaves, flowers in slightly scented panicles with pink petals, pod-shaped fruits with brown wings and yellowish-brown seeds (5 mm long, and 3 mm width) [51]. *P. piscipula* is found from Tamaulipas to the Yucatan Peninsula, Mexico, the United States (South Florida), and Honduras [52]. In the Yucatan Peninsula, *P. piscipula* is especially abundant in the secondary vegetation of medium sub-evergreen forests and medium sub-deciduous forests [48]. Beekeepers value this tree species because it remains in bloom for four months. The region’s inhabitants use their leaves to make medicines and as a flavoring herb for a famous dish from the Yucatan Peninsula known as “*Cochinita pibil*”. Its wood is of excellent quality for construction and combustion [53]. In some parts of the Yucatan Peninsula is known as the mother of the candle. Its flowering season goes from January to April, followed by a fruiting season between February and June [52].

Within the Yucatan Peninsula, the Campeche state is considered a tropical zone [49], with forest, savannah, coast, and sea, with the forest predominating, which covers 80% of the territory [54]. The tropical forest vegetation is high-evergreen and semi-evergreen forests, medium deciduous and semi-evergreen forests, and low deciduous and semi-evergreen forests [54]. According to Flores and Espejel [49], in San Francisco de Campeche, the capital city of the Campeche state, the dominant vegetation is the semi-deciduous forest, especially in the north, center, and a little to the south. The Fabaceae and Rubiaceae species are representative of the floristic composition in this city [55]. This plant community in areas surrounding of the San Francisco de Campeche is dominated, in addition to other species such as *P. piscipula* and *E. cyclocarpum* trees [49]. The annual average temperature is 26 °C, with maximum levels before the summer solstice at an average of 28 °C, reaching a historical maximum temperature of 52 °C. The rainy season is between June to October (mean precipitation of 1634.5 mm [56] and the dry season (absence of rain) is from January to mid-May [57]. The highest relative humidity is during September (78.6%), and the lowest is in April (55.6%) [58].

### 4.2. Seed Collection

We collected seeds of at least ten mother plants of both *E. cyclocarpum* and *P. piscipula* from the surroundings of the city of San Francisco de Campeche (Altitude: 1 m., Latitude: 19°51′00″ N, Longitude: 90°31′59″ W), municipality of Campeche, Mexico, from April to June of 2021. The harvested seeds of both species were stored under standard conditions in plastic and airtight bags (one bag per each mother plant of each species) and kept at room temperature (25 °C ± 2 °C and 60–80% relative humidity) in normal day/night conditions [59,60] until experimentation in September 2021.

### 4.3. Seed Disinfection

Before the start of the experiment, all harvested seeds of *E. cyclocarpum* and *P. piscipula* were placed in a 20% commercial chlorine solution and shaken on a hot plate with a stirrer by two minutes. They were then placed in 70% ethyl alcohol and stirred on a hot plate with a stirrer. Finally, they were given three washes with distilled water [2].

### 4.4. Viability Test

We used the tetrazolium (2,3,5-tryphenil tetrazolium chloride, TTC) test to evaluate seed viability. TTC-test is a rapid method, commonly used to assess the seed viability [61]. TTC-test is normally determined by a topographical method (visual observation) in order to characterize the pattern and intensity of staining and coloration in individual seed embryos [62,63]. Thus, TTC-test has been suggested as reliable as germination tests in several plant species, see [63], and a positive correlation between viability and germination of seeds is expected [64].

We placed 50 seeds of each species in five groups of ten. To facilitate the entry of the solution into the seed, we made an incision with a scalpel parallel to the micropyle axis. We placed seeds from each group in a beaker and soaked them with 20 mL of distilled water for 24 h before being placed in the tetrazolium solution. Subsequently, we removed distilled water from each baker and added 20 mL tetrazolium solutions at 1%. Each baker was covered and wrapped with aluminum foil to maintain the seeds in darkness at 25 °C for 48 h. Subsequently, we observed under a stereomicroscope. We considered viable seeds when showing their cotyledons and embryos red-stained without apparent damage [65]. Viability was estimated for each species and presented as a percentage (%) [59,60].

### 4.5. Imbibition Rate

To evaluate the water permeability into seeds of *E. cyclocarpum* and *P. piscipula*, we assessed the seed imbibition in terms of fresh weight gain through time in both species [11]. We used 50 random seeds of each species (without scarification) and grouped them into five replicates of ten seeds. We placed them in plastic containers (one seed per container) with distilled water. Because the size of the seeds from the two studied species is different, we used different kinds of containers for each species. For *E. cyclocarpum*, we use hermetic plastic bottles (20 mL), and for *P. piscipula*, we use Eppendorf tubes (2 mL). We recorded the initial fresh weight value of each seed for each species before being placed in containers. We kept the *E. cyclocarpum* and *P. piscipula* seeds at 25 °C and registered the fresh weight values every 24 h for each replicate and species during three uninterrupted days (72 h). We considered imbibed seeds when they showed an increase in fresh weight by at least 100% [60]. With these data, we obtained the number of imbibed seeds (%), as well as the imbibition rate (%) for each species.

### 4.6. Effect of Pre-Germinative Treatments in Fresh Seeds

We evaluated the effect of different pre-germination treatments in fresh seeds of *E. cyclocarpum* and *P*. *piscipula* to break PY. We applied a mechanical scarification treatment. For this, we cut with a scalpel in the region opposite the micropyle of the seeds [22]. We also applied thermal scarifications by dipping seeds in boiling water for 5 s, 10 s, and 15 s. Here, seeds were placed inside a stainless-steel tea infuser of 50 mm of diameter and posteriorly dipping into a beaker containing 150 mL of distilled water [2]. We also used intact seeds (not scarification) as a control treatment. We used a completely random design, with ten replicates of 10 seeds for each treatment and species. We placed seeds in Petri dishes with 17.5 mL of distilled water using a sterile cotton layer as a substrate and sealed them with parafilm (Parafilm M, Pechiney Plastic Packaging, Chicago, IL, USA). We kept the Petri dishes in a temperature room at 25 °C and 60–80% relative humidity, with a photoperiod of 12 h [59]. We recorded the number of germinated seeds daily for 30 days, and with these data, we determined the daily germination and final germination percentages in each treatment and species.

### 4.7. Morphological Characterization of the Seed Coat

For both *E. cyclocarpum* and *P. piscipula*, a sample of six seeds were used. We used three complete seeds of both species to characterize the external section. Additionality, other three seeds were longitudinal sectioned with a scalpel to characterize the internal section. Both complete and sectioned seeds were mounted on carbon double-sided adhesive tape on metal pins. We analyzed the external and internal structures of the seed coat [2] with a scanning electron observed (Scanning Electron Microscope FlexSEM 1000 Hitachi) at 20 kilovolts (kV), and an angle of inclination of 0°. There was no need for gold coating for any of the samples before the SEM analysis.

### 4.8. Statistical Analysis

Seed viability values were analyzed using a one-way ANOVA considering the species as a predictive factor. Percentage of imbibed seeds (fresh weight) were also analyzed using a one-way ANOVA considering time as a predictive factor. In this analysis, we do not consider the species as a predictive factor due to the intrinsic morphological differences in fresh weight between *P. piscipul*a and *E. cyclocarpum* described above. To analyze imbibition rate in both species we applied a repeated measures of ANOVA considering species and time as predictor factors. In the case of daily germination in *P. piscipul*a and *E. cyclocarpum*, we also applied a repeated measure of ANOVA, taking into account species, treatments, and time as predictor factors. We also analyzed the values of the final germination considering as predictive factors the species and the pre-germination treatments using a factorial ANOVA. We analyzed the data on the structure of seed coats descriptively [2]. The normality of all the quantitative data and the homoscedasticity of the residuals in all the quantitative variables were corroborated. We analyzed all quantitative analyses with IBM SAS Statistics (version 9.4, for windows).

## 5. Conclusions

The *E. cyclocarpum* and *P. piscipula* seeds show a high water-impermeability and both a low proportion of imbibed seeds and imbibition rate, limiting the germination process from naturally occurring. The results of the morphological characterization of the seed coat in both species corroborated the presence of PY because they showed morphological structures such as macrosclereids cells, spongy parenchyma, and palisade parenchyma, which are considered direct evidence of the existence of PY in Fabaceae species. Finally, the seeds from both species became permeable and displayed high germination by application of pre-germinative treatments. *E. cyclocarpum* had the highest germination percentage in the mechanical scarification, while *P. piscipula* showed the highest germination percentage in thermal shocks with boiling water for 10 s. The present research seeks to promote the sustainable use of *E. cyclocarpum* and *P. piscipula* and contribute to the creation of effective conservation strategies and enhance their biotechnological applications.

## Figures and Tables

**Figure 1 plants-11-02844-f001:**
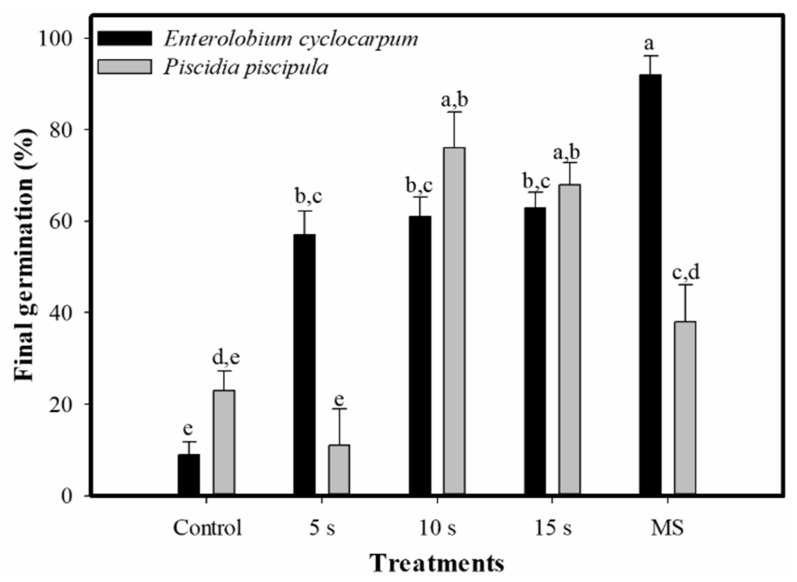
Final germination percentage (mean ± standard error, n = 10) per treatment (boiling water for 5 s, 10 s and 15 s and MS = mechanical scarification) in the studied species. Different letters indicate significant differences between treatments and species.

**Figure 2 plants-11-02844-f002:**
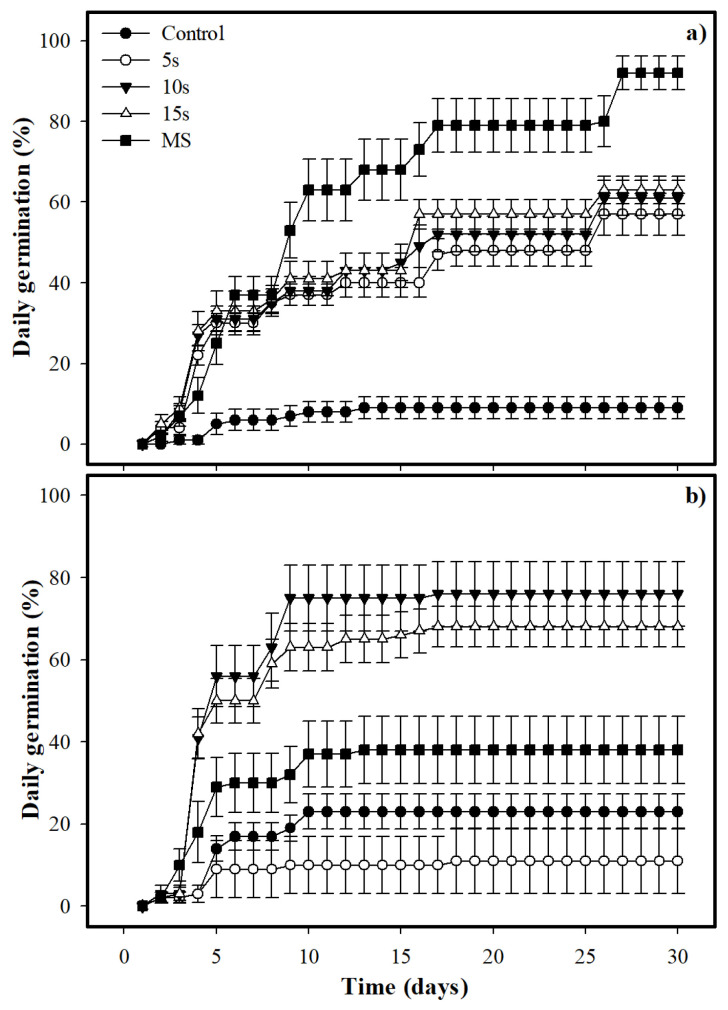
Daily seed germination (mean ± standard error, n = 10) in (**a**) *Enterolobium cyclocarpum* and (**b**) *Piscidia piscipula* in different germinative pre-treatment (boiling water for 5 s, 10 s and 15 s and MS = mechanical scarification).

**Figure 3 plants-11-02844-f003:**
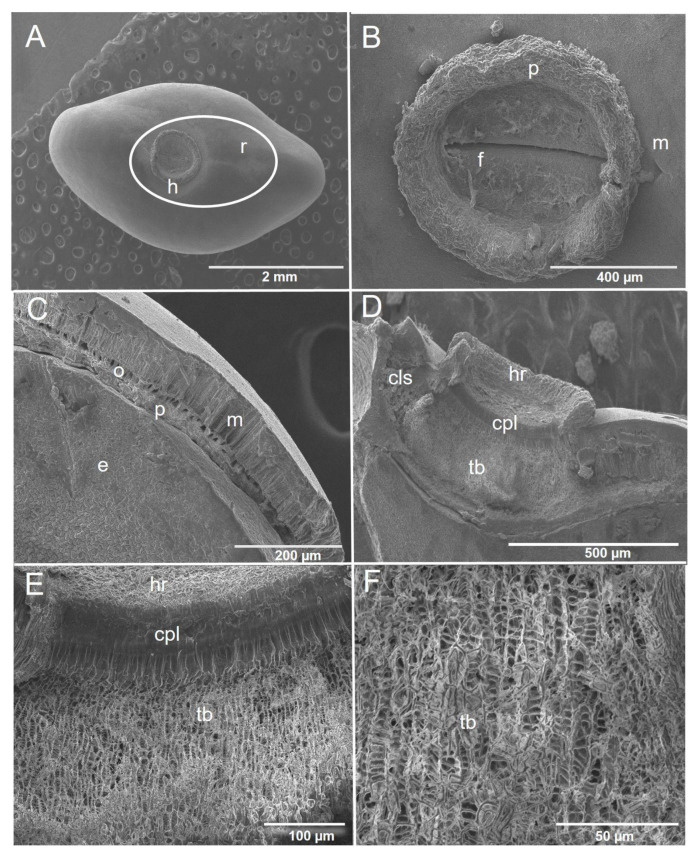
Morphological characterization of the *Piscidia piscipula* seed coat. (**A**) External section, h, hilum, r, raphe, (**B**) Detail of the hilar region: f, hilar fissure, p, funicular parenchyma, m, micropyle, (**C**) Longitudinal section of the epidermis: e, embryo, m, macrosclereids, o, osteosclereids, p, parenchyma, (**D**) Longitudinal section of the hilar region: cls, cushion-like structure, cpl, counter-palisade layer, hr, hilar region, tb, tracheid bar, (**E**) Detail of interior of hilar region, cpl, counter-palisade layer, hr, hilar region, tb, tracheid bar, and (**F**) Approach to the tracheid bar cell, tb, tracheid bar.

**Figure 4 plants-11-02844-f004:**
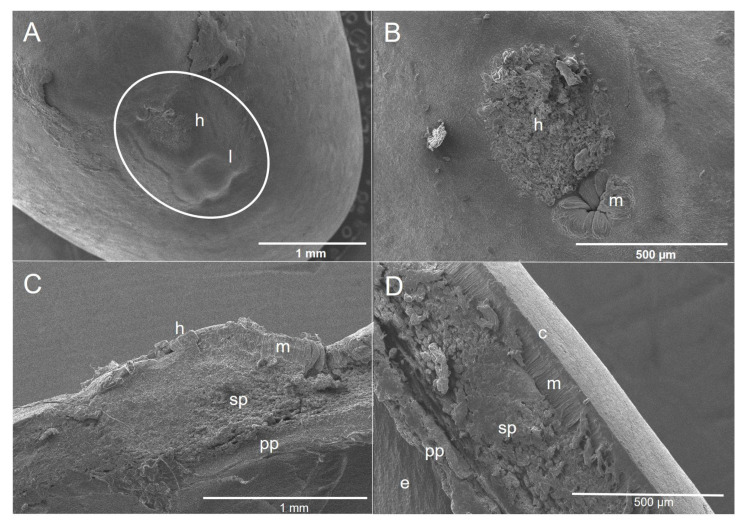
Morphological characterization of the *Enterolobium cyclocarpum* seed coat. (**A**) External section, h, hilum, l, lens (**B**) Detail of the hilar region: h, hilum, m, micropyle, (**C**) Longitudinal section of the hilar region: h, hilum, m, macrosclereids, sp, spongy parenchyma, pp, palisade parenchyma, (**D**) Longitudinal section of the epidermis: c, cuticle, e, embryo, m, macrosclereids, sp, spongy parenchyma, pp, palisade parenchyma.

## Data Availability

Not applicable.

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
