# Peer review of "Pre-Germinative Treatments and Morphophysiological Traits in Enterolobium cyclocarpum and Piscidia piscipula (Fabaceae) from the Yucatan Peninsula, Mexico"

_plants, 2022, doi:10.3390/plants11212844_

Round 1

Reviewer 1 Report

Seed dormancy is an important problem during seed germination.The seeds of Enterolobium cyclocarpum and Piscidia piscipula,which are two important tree species distributed in  Mexico, were used to evaluate the effect of seed pre-germination treatments and to characterize the structures involved on the presence of physical dormancy (PY). The results are certain interested. But in general, the results are too simple and preliminary.

1. Why do the two methods used by the author improve the seed germination rate? The author should at least show physiological datas to explain the mechanism ?

2.The author used scanning electron microscope to analyze the surface structure of two kinds of seeds. What is the relationship between this structure and seed germination? The current results do not show that the seed surface structure will inevitably lead to seed dormancy

Author Response

Comments and Suggestions for Authors

Seed dormancy is an important problem during seed germination. The seeds of Enterolobium cyclocarpum and Piscidia piscipula, which are two important tree species distributed in  Mexico, were used to evaluate the effect of seed pre-germination treatments and to characterize the structures involved on the presence of physical dormancy (PY). The results are certain interested. But in general, the results are too simple and preliminary.

  1. Why do the two methods used by the author improve the seed germination rate? The author should at least show physiological data to explain the mechanism ?

Our response

We added information in discussion section to better interpret these results as follows:

“It has been documented that the seed coat functions as regulator in imbibition phase [44] and a high and fast germination are inversely correlated with high seed coat hardness [45]. Therefore, when the seed coat is scarified, it can potentially decrease mechanical resistance to germination [46]. Thus, the thermal shocks with boiling water for 10 s (P. piscipula) and mechanic scarification (E. cyclocarpum) treatments influenced the breaking of PY, increasing the water uptake, and consequently high germination in both species.

The effect of temperature on the natural decomposition of impermeable seed coats may be through the heating effect of solar radiation on the surface layers of dry and/or moist soils, together with night cooling, resulting in a combination of exposure to temperature fluctuations, thus favoring germination to take place [2]. In the case of P. piscipula, the higher germinative efficiency promoted for heat shocks by 10 s possibly favored the formation of cracks on the seeds allowing a more effective separation of the macrosclereids cells as well as of parenchymal tissue. It is possible that mechanical scarification had high efficiency in terms of germination in E. cyclocarpum because seeds of this species displayed a high proportion of spongy parenchyma and palisade parenchyma than P. piscipula, and this treatment facilitated their removal to overcome this barrier.”

2.The author used scanning electron microscope to analyze the surface structure of two kinds of seeds. What is the relationship between this structure and seed germination? The current results do not show that the seed surface structure will inevitably lead to seed dormancy

Our response

We added information in the introduction and discussion sections to explain some morphological attributes of species with physical dormancy as follows:

“One of the most common types of dormancies in plant species is physical dormancy (PY), in which seed or fruit coats are water-impermeable and unable to imbibe water, limiting the germination process to be carried out [1]. The seeds or fruit coat are water-impermeable due to showing one or more layers of palisade cells called macrosclereids [4,2], as well as spongy parenchyma and palisade parenchyma [42]. In this study, both P. piscipula and E. cyclocarpum showed similar morphological structures; therefore, the presence of PY in both species was corroborated.”

Reviewer 2 Report

The topic considered is an interesting one. It showed us Pre-germination treatments and morphological characterization of the seed coat in two Fabaceae species from the Yucatan Peninsula, Mexico. But I think that the paper needed to be minor revised

1.     The harvested seeds of both species were stored in plastic and airtight bags and kept at room temperature (25 °C ± 2 °C and 60-80 % relative humidity) ….

Please confirm again the storage relative humidity may be too high

2.     Please confirm before the start of the experiment, the E. cyclocarpum and P. piscipula seeds were placed in a 20 % commercial chlorine solution (Reference ISTA method)

and shaken on a hot plate with a stirrer minutes.

3.     In this manuscript, Abstract – you mentioned 1) to examine the seed imbibition rate are also in the focus. But the title of the paper did not reflect that.

This sentence needs a reference. A little bit predictive as well as we can’t be sure water absorption level is the factor responsible for all the changes. EDIT: Did moisture content study in Figure 4

Maybe you can update by sentence to reflect this?

4.     This paper used a completely random design, with ten replicates of 10 seeds for each treatment and species. But Figure 4. "standard error" can not reflect that.

5.     Figure 4. – Did pre-treatment (boiling water) increase the water uptake as well as a result of the cracks? 

Author Response

Comments and Suggestions for Authors

The topic considered is an interesting one. It showed us Pre-germination treatments and morphological characterization of the seed coat in two Fabaceae species from the Yucatan Peninsula, Mexico. But I think that the paper needed to be minor revised

1.The harvested seeds of both species were stored in plastic and airtight bags and kept at room temperature (25 °C ± 2 °C and 60-80 % relative humidity) ….

Our response

We added complementary information as follows:

“The harvested seeds of both species were stored under standard conditions in plastic and airtight bags (one bag per each mother plant of each species) and kept at room temperature (25 °C ± 2 oC and 60-80 % relative humidity) in normal day/night conditions [60,61] until experimentation in September 2021.”

Please confirm again the storage relative humidity may be too high?

Our response

The relative humidity values are standard conditions for seed storage, see [60,61].

2.Please confirm before the start of the experiment, the E. cyclocarpum and P. piscipula seeds were placed in a 20 % commercial chlorine solution (Reference ISTA method) and shaken on a hot plate with a stirrer ? minutes.

Our response

We added the complementary information as follows:

“Before the start of the experiment, all harvested seeds of the E. cyclocarpum and P. piscipula seeds were placed in a 20 % commercial chlorine solution and shaken on a hot plate with a stirrer by two minutes. They were then placed in 70 % ethyl alcohol and stirred on a hot plate with a stirrer. Finally, they were given three washes with distilled water [2].”

3.In this manuscript, Abstract – you mentioned 1) to examine the seed imbibition rate are also in the focus. But the title of the paper did not reflect that.

Our response

We changed the manuscript title to include aspects such as seed imbibition rate as follows:

“Pre-germinative treatments and morphophysiological traits in Enterolobium cyclocarpum and Piscidia piscipula (Fabaceae) from the Yucatan Peninsula, Mexico”

This sentence needs a reference. A little bit predictive as well as we can’t be sure water absorption level is the factor responsible for all the changes. EDIT: Did moisture content study in Figure 4?

Maybe you can update by sentence to reflect this?

Our response

The humidity relative conditions used in the germination experiments were homogeneous and described in section 4.6 of the materials and methods as follows:

“We kept the Petri dishes in a temperature room at 25 °C and 60-80% relative humidity, with a photoperiod of 12 h [60].”

4.This paper used a completely random design, with ten replicates of 10 seeds for each treatment and species. But Figure 4. "Standard error" cannot reflect that.

Our response

We added the requested information as follows:

“Figure 4. Daily seed germination (mean ± standard error, n = 10) in a) Enterolobium cyclocarpum and 2) Piscidia piscipula in different germinative pre-treatment (boiling water for 5 s, 10 s and 15 s and MS = mechanical scarification).”

5.Figure 4. – Did pre-treatment (boiling water) increase the water uptake as well as a result of the cracks? 

Our response

We rewrote the interpretation of these results in the discussion section as follows:

“In the case of P. piscipula, the higher germinative efficiency at promoted for thermal heat shocks by 10 s possibly favored the formation of cracks on the seeds allowing a more effective separation of the macrosclereids cells as well as of parenchymal tissue.”

Reviewer 3 Report

The paper is good and the subject and the assessment is really interesting for readers involved in seed science and practice.

The reviewer would like to clear three points that should be interpreted or highlighted a litte bit more succesfully.

1. The outcome of the research is promising, however the results should be given a more clear interpretation - namely what do the authors think about the future appllicability of their findings.

2. In the materials and methods the authors describe the viability tests. As we know - TTC or any other chemical indicators give no direct result in comparison with germination tests and/or vigour tests - however such results may have high correlation with the latter. It would be beneficial to highlight this part of the methodology in more details, and supporting that with some more former results from literature.

3. The descrition of scarification is very poor. A more detailed methodological description would make a benefit.

Author Response

Comments and Suggestions for Authors

The paper is good, and the subject and the assessment is really interesting for readers involved in seed science and practice.

The reviewer would like to clear three points that should be interpreted or highlighted a litte bit more successfully.

  1. The outcome of the research is promising; however, the results should be given a clearer interpretation - namely what do the authors think about the future applicability of their findings.

Our response

We expand the perspectives of this research in the discussion section as follows:

“In future works, it is important to characterize the initial site of water entry in the P. piscipula and E. cyclocarpum seeds; in addition, the possible external changes of seed coat after application of pre-germinative treatments and the effect of some microorganisms such as fungi species, which also could help to break the PY in these species. An aspect that has been almost neglected is the dynamics of the soil seed bank in both species, see [47, 48]. Thus, studies of soil seed bank dynamics in E. cyclocarpum and P. piscipula are also crucial in future research.

  1. In the materials and methods the authors describe the viability tests. As we know - TTC or any other chemical indicators give no direct result in comparison with germination tests and/or vigor tests - however such results may have high correlation with the latter. It would be beneficial to highlight this part of the methodology in more details and supporting that with some more former results from literature.

Our response

We have expanded the description of the viability test in the materials and methods section as follows:

“We used the tetrazolium (2,3,5-tryphenil tetrazolium chloride, TTC) test to evaluate seed viability. TTC-test is a rapid method, commonly used to assess the seed viability [62]. TTC-test is normally determined by a topographical method (visual observation) in order to characterize the pattern and intensity of staining and coloration in individual seed embryos [63,64]. Thus, TTC-test has been suggested as reliable as germination tests in several plant species, see [64], and a positive correlation between viability and germination of seeds is expected [65].”

In addition, we associated the viability results found here with the germination outcomes in discussion as follows:

“We found high the seed viability values in these species and consequently, the E. cyclocarpum and P. piscipula seeds became permeable and showed a high seed germination through the application of pre-germination treatments.”

  1. The description of scarification is very poor. A more detailed methodological description would make a benefit.

Our response

We have expanded the description of the scarification methods in the materials and methods section as follows:

We applied a mechanical scarification treatment. For this, we cut with a scalpel in the region opposite the micropyle of the seeds [23]. We also applied thermal scarifications by dipping seeds in boiling water for 5 s, 10 s, and 15 s. Here, seeds were placed inside a stainless-steel tea infuser of 50 mm of diameter and posteriorly dipping into a beaker containing 150 ml of distilled water [2].”

Reviewer 4 Report

This manuscript describes analysis of two legume species from Mexico, study of treatments leading to release of dormancy. It is know that legumes have mostly physical dormancy type, executed by seed coat permeability. It is novel study for given species.

I have several specific comments:

In abstract:  last 2 sentences should be modified. Presence of macrosclereids does not automatically mean that there is PY dormancy. The decissive point is the water permeability and imbibition rate. I do now agree that study promotes sustainable use of species, it contributes to better understading of seed biology and might help in propagation.

Introduction: very substantial part is very general ans should be omitted. There are know facts which might be reduced to only few sentences, while there would be usefull to provide examples from relevant legume studies - which will be later on recapitulated in discussion.

Do now write "we had two hypothesis..." instead Following hypothesis were tested in the study....

similarly, be specific - do not say "aim of the study was to evaluate some treatments.." but write which ones

Minor points: Fig. 1 can be omitted as it is enough to write viability values in text. Similarly Fig 2 on fresh weight and imbibition rate.

the most important ones are data on percentage of imbibition (Fig. 3 - which should become figure 1)

In Fig. 4 please remove  label "d" from values on days in axis X, it is enought to say 1-2-3 etc. (Time (days)

Discussion part:

You have tested rather artificial treatments (boiling water and mechanical scarification) could you elaborate on factors acting in natural conditions ? Temperature/humidity oscilations, microbial decay of seed coat etc. Taking examples from published studies on similar species.

Is there is soil seed bank in given species ? How persistent are seeds in soil?

Author Response

Comments and Suggestions for Authors

This manuscript describes analysis of two legume species from Mexico, study of treatments leading to release of dormancy. It is known that legumes have mostly physical dormancy type, executed by seed coat permeability. It is novel study for given species.

I have several specific comments:

In abstract:  last 2 sentences should be modified. Presence of macrosclereids does not automatically mean that there is PY dormancy. The decissive point is the water permeability and imbibition rate. I do now agree that study promotes sustainable use of species, it contributes to better understading of seed biology and might help in propagation.

Our response

We rewrote the last part of the abstract as follows:

The presence of PY was confirmed in both species because they showed low seed permeability, and imbibition rate; furthermore, exhibited macrosclereids cells.

Introduction: very substantial part is very general ans should be omitted. There are know facts which might be reduced to only few sentences, while there would be usefull to provide examples from relevant legume studies - which will be later on recapitulated in discussion.

Our response:

We have improved the wording of the introduction for a better understanding.

Do now write "we had two hypotheses..." instead Following hypothesis were tested in the study....

similarly, be specific - do not say "aim of the study was to evaluate some treatments.." but write which ones

Our response

We rewrote the hypotheses and aims as follows:

“Therefore, following hypotheses were tested in this study: 1) the seeds of E. cyclocarpum and P. piscipula will display PY, and they will increase the seed germination by applying pre-germination treatments, and 2) the presence of PY in both species is because they show water-impermeability and low imbibition rate due to the presence of specialized cells known as macrosclereids, and structures such as water gap or cushion-like structure are involved in the process of seed imbibition in these species. Thus, we evaluated the efficiency of some seed pre-germination treatments in E. cyclocarpum and P. piscipula; in addition, we examined the seed permeability and imbibition rate in both species, and we characterized the external and inner morphology of the testa in the E. cyclocarpum and P. piscipula seeds to describe the structures involved on the presence of PY in these species.”

Minor points: Fig. 1 can be omitted as it is enough to write viability values in text.

Our response

We delete Figure 1 as suggested by revisor.

Similarly, Fig 2 on fresh weight and imbibition rate.

the most important ones are data on percentage of imbibition (Fig. 3 - which should become figure 1)

Our response

We delete Figure 2 as suggested by revisor.

In Fig. 4 please remove  label "d" from values on days in axis X, it is enought to say 1-2-3 etc. (Time (days)

Our response

We have made the change requested by the reviewer in this figure.

Discussion part:

You have tested rather artificial treatments (boiling water and mechanical scarification) could you elaborate on factors acting in natural conditions ? Temperature/humidity oscilations, microbial decay of seed coat etc. Taking examples from published studies on similar species.

Our response

We added information in discussion section to better interpret these results as follows:

“It has been documented that the seed coat functions as regulator in imbibition phase [44] and a high and fast germination are inversely correlated with high seed coat hardness [45]. Therefore, when the seed coat is scarified, it can potentially decrease mechanical resistance to germination [46]. Thus, the thermal shocks with boiling water for 10 s (P. piscipula) and mechanic scarification (E. cyclocarpum) treatments influenced the breaking of PY, increasing the water uptake, and consequently high germination in both species.

The effect of temperature on the natural decomposition of impermeable seed coats may be through the heating effect of solar radiation on the surface layers of dry and/or moist soils, together with night cooling, resulting in a combination of exposure to temperature fluctuations, thus favoring germination to take place [2]. In the case of P. piscipula, the higher germinative efficiency promoted for heat shocks by 10 s possibly favored the formation of cracks on the seeds allowing a more effective separation of the macrosclereids cells as well as of parenchymal tissue. It is possible that mechanical scarification had high efficiency in terms of germination in E. cyclocarpum because seeds of this species displayed a high proportion of spongy parenchyma and palisade parenchyma than P. piscipula, and this treatment facilitated their removal to overcome this barrier.”

Is there is soil seed bank in given species ? How persistent are seeds in soil?

Our response:

We added information about soil seed bank dynamics of both study species in discussion as follows:

An aspect that has been almost neglected is the dynamics of the soil seed bank in both species, see [47,48]. Thus, studies of soil seed bank dynamics in E. cyclocarpum and P. piscipula are also crucial in future research.

Reviewer 5 Report

This work by Arceo-Gomez and colleagues investigates germination features of two Fabaceae species from Mexico.

While the potential interest of the paper is naturally limited by its niche object of study, the work provides enough scientific soundness to be considered for publication in Plants.

Some minor suggestions:

- I would suggest to include the name of the species in the title;

- Bibliography needs major review. Inappropriate references are used in the introduction. For example [1] and [3] should be replaced with sources available also in English, in order not to limit their access to broader audiences. [2] is not a peer-reviewed work. DOI are missing for most of the cited references, as also precise details when books are cited.

- Most of the numerical results describing the data are currently included in the paragraphs, severely limiting their readability. They can be all included in simpler and clearer tables.

Author Response

Comments and Suggestions for Authors

This work by Arceo-Gomez and colleagues investigates germination features of two Fabaceae species from Mexico.

While the potential interest of the paper is naturally limited by its niche object of study, the work provides enough scientific soundness to be considered for publication in Plants.

Some minor suggestions:

- I would suggest including the name of the species in the title;

Our response

We changed the manuscript title to include aspects such as seed imbibition rate as follows:

“Pre-germinative treatments and morphophysiological traits in Enterolobium cyclocarpum and Piscidia piscipula (Fabaceae) from the Yucatan Peninsula, Mexico”

- Bibliography needs major review. Inappropriate references are used in the introduction. For example [1] and [3] should be replaced with sources available also in English, in order not to limit their access to broader audiences. [2] is not a peer-reviewed work. DOI are missing for most of the cited references, as also precise details when books are cited.

Our response

We have updated the cited references, removed 1-3 references and added the DOI of all the references that contain it, following editorial rules of the journal.

- Most of the numerical results describing the data are currently included in the paragraphs, severely limiting their readability. They can be all included in simpler and clearer tables.

Our response

We have improved the wording of the results for a better understanding.

Round 2

Reviewer 1 Report

After modification, the manuscript has significantly improved in terms of language and scientific description, meeting the publishing requirements. However, the data and quality of the paper are average.
1. How many seeds are used by scanning electron microscope, and whether the results are common in the seeds?

Author Response

How many seeds are used by scanning electron microscope?

Our response

We rewrote part of the morphological characterization of the seed coat in materials and methods as follows:

“For both E. cyclocarpum and P. piscipula, a sample of six seeds were used. We used three complete seeds of both species to characterize the external section. Additionality, other three seeds were longitudinal sectioned with a scalpel to characterize the internal section.”

and whether the results are common in the seeds?

Our response

In discussion, we included information to discuss the results in terms of the morphological characterization of the seed coat found in this study

“The presence of macrosclereids and osteosclereids cells [2,42] and cushion-like structure and tracheid bars [2,5,6] inside seeds are considered as direct evidence of the presence of PY in Fabaceae species. In this sense, it has been suggested that the P. piscipula [12] and E. cyclocarpum [20] seeds show PY.

The presence of a cushion-like structure and tracheid bar were only corroborated in P. piscipula. Cushion-like structure has been suggested to be found at Faboideae, a subfamily within Fabaceae, which groups P. piscipula [43]. E cyclocarpum is found into Mimosoideae subfamily. It is possible that this is the explanation why cushion-like structure was not found in this species. Although we found almost all these morphological structures described above, our results about morphological characterization also confirm the assumption of the existence of PY in E. cyclocarpum and P. piscipula [12,20].”